# Perceptual cognitive abilities in young athletes: A gender comparison

**Isabelle Legault**[1]*, **Dylan Sutterlin-Guindon**[1], **Jocelyn Faubert**[2]

**1** Collège Lionel-Groulx, Ste-Thérèse, Qc, Canada, **2** Faubert Lab, School of Optometry, Université de Montréal, Montreal, Qc, Canada

* isabelle.legault@clg.qc.ca

**Data Availability Statement:** All relevant data are within the article and its Supporting Information files.

**Funding:** IL received a grant from FRQS (Fonds de recherche du Québec - Santé #262061) https://frq.

## Abstract

To achieve optimal performance in sports, it is essential to have strong perceptual cognitive abilities. Evidence suggests that athletes have superior perceptual abilities compared to nonathletes. However, gender differences in athletes' perceptual cognitive abilities have not been previously reported. This project aims to evaluate perceptual cognitive abilities among male and female adolescents. To measure perceptual abilities, a 3-dimensional multiple-object tracking task was used. The findings confirm the superior perceptual cognitive abilities in young athletes relative to nonathletes. However, our results indicate differences in performance patterns between male and female athletes. These results demonstrate that sports engagement and perceptual cognitive abilities are strongly related during adolescence and that this relationship seems more prevalent in male athletes for this age group.

## Introduction

The adolescent period (12–20 years old) is characterized by puberty, hormonal changes and brain modifications and a greater propensity for taking risks. This is a period when teenagers experiment, engage in suboptimal decisions and actions such as unprotected sexual behavior, consume alcohol and drugs, and have dangerous driving behaviors [1–6]. The brain's neural maturation has not yet reached its peak during adolescence, which could be part of the cause. The human brain develops asymmetrically. The neural regions responsible for the treatment of emotions are functional from the beginning of adolescence, while the parts of the prefrontal cortex (PFC) responsible for decision-making and information processing reach maturation at approximately age 25 [5,7,8]. Teenager decision-making is dependent on modifications observed in the PFC, limbic system and dopaminergic (DA) system [9,10]. Changes in the PFC during adolescence lead to DA imbalances, and it has been hypothesized that pleasure is driven by short-term rewards; teenagers continuously seek novelty and rewarding stimuli [5]. Furthermore, gender differences have been observed during the myelination process of the PFC involved in the management of risky situations and executive function. This process starts earlier in the female brain than in the male brain [11].

Vaughan and colleagues [12] found that a higher quality of decision-making was associated with higher athletic expertise. Athletes tend to exhibit better anticipation of emotions, which influences their decision-making. Furthermore, it seems imperative that to achieve high

gouv.qc.ca/ The funders had no role in study design, data collection and analysis, decision to publish, or preparation of the manuscript.

**Competing interests:** One of the authors is director of Faubertlab at the University of Montreal and he is the Chief Science Officer of Cognisens Athletics Inc. who produces the commercial version of the NeuroTracker used in this study. In this capacity, he holds shares in the company. This does not alter our adherence to your journal policies on sharing data and materials.

performance in sports, an athlete must learn how to properly manage risks, act quickly and have good perceptual cognitive abilities. Athlete expertise is linked to decision-making in high-pressure situations and stressful environments [13,14]. In team sports such as soccer, the player must constantly pay attention to the surrounding environment to make good decisions. For example, the player needs to evaluate the position of other players on the field and visualize the direction of the ball. It is often said that the best players in the world have a "vision of the game", which is out of the ordinary for most players [15]. Researchers have found that athletes have quicker cognitive processing abilities than nonathletes [16,17].

Expertise in information processing is associated with skills that are scientifically termed "perceptual cognitive" abilities. Perceptual cognitive skills refer to the ability to simultaneously integrate external stimuli and then use them to act correctly within the demands of a specific task [18]. Meta-analyses reveal that sports experts perform better than nonexperts in certain aspects of perceptual cognitive skills. When detecting perceptual cues, experts exhibit increased accuracy and have a shorter response time. Additionally, in visual search tasks, experts show shorter eye fixations, which suggests that they have quicker visual information integration [19]. They also exhibit better performance in terms of speed processing, reaction time, divided attention, focused attention tasks and better integrated information in their visual field [17,20]. Moreover, regarding perceptual cognitive skills, concepts of anticipation and decision-making have been reported to be critical components and predictors of elite performance [20–22].

The development of these attentional and perceptual abilities changes with age and can be modified with sports experience [15,21,23–25]. To measure this perceptual cognitive ability, a 3-dimensional multiple-object tracking (3D-MOT) task consisting of simultaneously tracking several moving objects among moving distractors can be used. 3D-MOT requires controlled endogenous processes consisting of selecting certain objects (targets) and ignoring others (distractors). Attention and the ability to handle multiple objects at a time are required for the execution of many daily tasks. This capacity can be evaluated by measuring the number of items tracked without error [26–28] or the speed at which one can track a certain number of items [15,23,29]. It has been repeatedly demonstrated that the moving object tracking test is a complex task that involves attentional mechanisms, including selective attention [30–32] and sustained attention [33,34]. The 3D-MOT task includes a multiple-object training component, a large stimulated visual field area, and stereoscopic 3D information, and it allows for the measurement of speed thresholds. Faubert and Sidebottom [15] provided a detailed discussion of the underlying rationale for the importance of each element that composes the 3D-MOT speed training task. Additionally, this task has been measured in several populations, such as elderly individuals, children and athletes [15,21,23,25,35,36]. Faubert [23] observed that the MOT learning function was higher in athletes (professionals and elite amateurs) than in university students and that this function was dependent on sports performance level (professionals better than elite amateurs).

Although the level of the athlete is associated with better perceptual cognitive abilities, where neural changes are observed in functions required during the training such as attention (divided / sustained or selective) [37],it remains unclear whether there is a gender difference in both male and female athletes cognitive abilities. Voss et al. (2009) meta-analytic review suggested that male athletes observed a greater cognitive benefit than female athletes compared to the non-athletes control group but suggested that more studies recruiting both male and female is needed to permit more accurate data interpretation [17].

Moreover, there is a lack of literature on gender differences in the evaluation of perceptual cognitive ability tasks. A number of studies show gender differences across life span on spatial cognition [38–41]. More relevant to the present study, Roudaia and Faubert [29] supports a

significant gender difference in attentional tracking. Their results showed gender differences in tracking performance for younger and older adults, where males obtained higher tracking speed, which indicates a higher temporal resolution of attention. Furthermore, researchers have found a lower reaction time in females, which can be explained by their lower ability in attentional function [37–40] and revealed that females are more negatively influenced by distractors [41,42].

The purpose of the present study was therefore to establish whether gender differences are observed for tracking multiple moving objects in athlete and non-athlete populations.

## Materials & methods

### Participants

Forty athletes (21 boys and 19 girls) and forty nonathletes (20 boys and 20 girls) (μ 18.3 years old, range: 17–24 years old) participated in this study. Our sample size for 2 groups with a Power 1- β = .8 and Type I error rate, α = 0.05, was estimated at 63 participants. All participants were naïve to the purpose of the experiment. All subjects had normal or corrected-to-normal vision (6/6 or better) with normal stereoacuity as measured by the RANDOT Stereo Test (50 seconds of arc or better) [42]. Viewing was binocular. All participants completed the IVA-2 test, a screening measure for attention deficit disorder, and all scores were within the normal range. All participants had a history of fewer than two concussions. All athlete participants were members of a college sports team (collective sports such as soccer, football, or hockey) and played at least twice a week. Nonathlete participants were not involved in a regular sports practice (less than once a week), and they were not part of any sports teams within the last five years. Participants were recruited from the college population, and they were, for the majority, in a pre-university program. Ethical approval was obtained from the College Lionel Groulx Ethics Board (#2018–02), and written consent was obtained from all participants. The ethic board has authorized our participant (aged 16 years old and over) to give their own consent, based on the low risk factor in this study.

### Procedure

Participants completed a short questionnaire about their sports practice. Then, they proceeded with the 3D-MOT task. Participants were asked to wear anaglyph goggles, which allowed them to perceive the 3D characteristics of the image projected on a computer screen. The participants were seated at 30 cm from the screen and asked to stare at the fixation point, located straight ahead. Stimuli consisted of height spheres projected into a virtual cube measuring 324 cm$^2$. The anterior side of the cube measured 33.4˚ of visual angle. Three trials that lasted approximately 20 minutes were presented.

### Measurement

**3D-MOT paradigm.** The commercial version of the 3D-MOT speed threshold task, called NeuroTracker$^{TM}$ (CogniSens Inc.), was used to assess the perceptual cognitive abilities of the participants. The task was presented on a computer screen. Participants wore anaglyph glasses that allowed for 3D visualization of the stimuli. The observers ran 3 consecutive trials, with each session lasting approximately 8 minutes. The basic 3D-MOT trial sequence is presented in Fig 1 and comprises four steps (see legend). The task began at a given speed, and at the end of the trial, if all four spheres were not correctly identified, the next trial decreased the speed. If the four spheres were correctly identified, then the next trial was faster. Twenty trials were presented to the participants, and the final speed threshold was calculated as the average of the last 4 reversals.

**Fig 1.** Five steps of the 3D-MOT task: (a) presentation phase, where 8 spheres are shown in a 3D volume space; (b) indexing phase, where 4 spheres (targets) change color (red) and are highlighted by a halo for 1 second; (c) movement phase, where the targets indexed in stage b return to their original form and color and all spheres move for 8 seconds crisscrossing and bouncing off of each other and the virtual 3D volume cube walls that are not otherwise visible; and (d) identification phase, where the spheres come to a halt and the observer has to identify the 4 spheres originally indexed in phase (b). The spheres are individually tagged with a number so the observer can report the number; (e) Feedback was given to the observer.

## Data analysis

The task measured speed thresholds obtained by each participant after 3 blocks of 20 trials (total of 60 trials per subject). At any given trial if the participant gets all targets correct the moving items speed up by 0.5 log units in the next trial. If the participant makes one or more mistakes in the recall, the next trial slows down by 0.5 log units. The speed threshold value of a given block consists of the mean of the speed for the last 4 reversals. First, the speed value used for statistical analysis for the entire session is the average of the 3 block thresholds obtained for each of the participants. We are going to compare, by a statistical analysis (split-plot ANOVA) whether differences remain between groups and evaluated with Post hoc Bonferonni correction if significant interactions are obtained.

Finally, we predict a different improvement (speed thresholds) within groups between block 1 and block 3. To perform this analysis, we will subtract the results obtained in the 3rd trial from those obtained in the first and we will perform a t-test on these results. A significant difference will tell us that there is a difference in the speed of learning in our participants between trials 1 and 3.

## Results

IBM SPSS statistics version 24 (SPSS, Chicago, Illinois) was used for all statistical analysis, with alpha set at $p \leq 0.05$. A split-plot ANOVA on speed thresholds comparing groups revealed a significant group effect, $F(3,80) = 6.789$ $p < 0.01$, $\eta^2 = 0.698$. Subsequently, Post hoc comparisons using the Bonferonni correction indicated that the mean score for the male athlete group ($M = 1.73$, $SD = 0.48$) was significantly different than female athletes and nonathlete males and females, see Table 1.

As observed in Fig 2, male athletes reached higher speed thresholds than other groups. Furthermore, a paired t-test was performed on normalized data to observe the progression of the participants' speed thresholds between the first and last trials. The NT task is sensitive to repetition, and participants improve their performance significantly trial after the trial [15,21,25]. Normalized data showed differences between female athletes and other groups, leading to a more pronounced performance progression in the female athlete group (Table 2).

## Discussion

The main goal of this study was to establish whether young athletes show a higher capacity to track multiple moving objects than nonathletes and whether differences exist between males and females.

We reported a significant difference in speed thresholds between groups, and athletes obtained higher speed thresholds than nonathletes. These results are generally consistent with previous 3D-MOT research showing better performance in athletes than in nonathletes [23]

**Table 1. Post hoc comparisons.**

| (I)groups | J (groups) | Mean difference (I-J) | Std. Error | Sig. |
|---|---|---|---|---|
| AF | AM | -,43373* | 0.12462 | 0.005 |
| | NAF | 0.07179 | 0.12609 | 1.000 |
| | NAM | -0.05516 | 0.12462 | 1.000 |
| **AM** | **AF** | **,43373*** | **0.12462** | **0.005** |
| | **NAF** | **,50552*** | **0.12297** | **0.001** |
| | **NAM** | **,37857*** | **0.12146** | **0.015** |
| NAF | AF | -0.07179 | 0.12609 | 1.000 |
| | AM | -,50552* | 0.12297 | 0.001 |
| | NAM | -0.12695 | 0.12297 | 1.000 |
| NAM | AF | 0.05516 | 0.12462 | 1.000 |
| | AM | -,37857* | 0.12146 | 0.015 |
| | NAF | 0.12695 | 0.12297 | 1.000 |

A: athletes / NA: non-athletes / M: male / F: female.

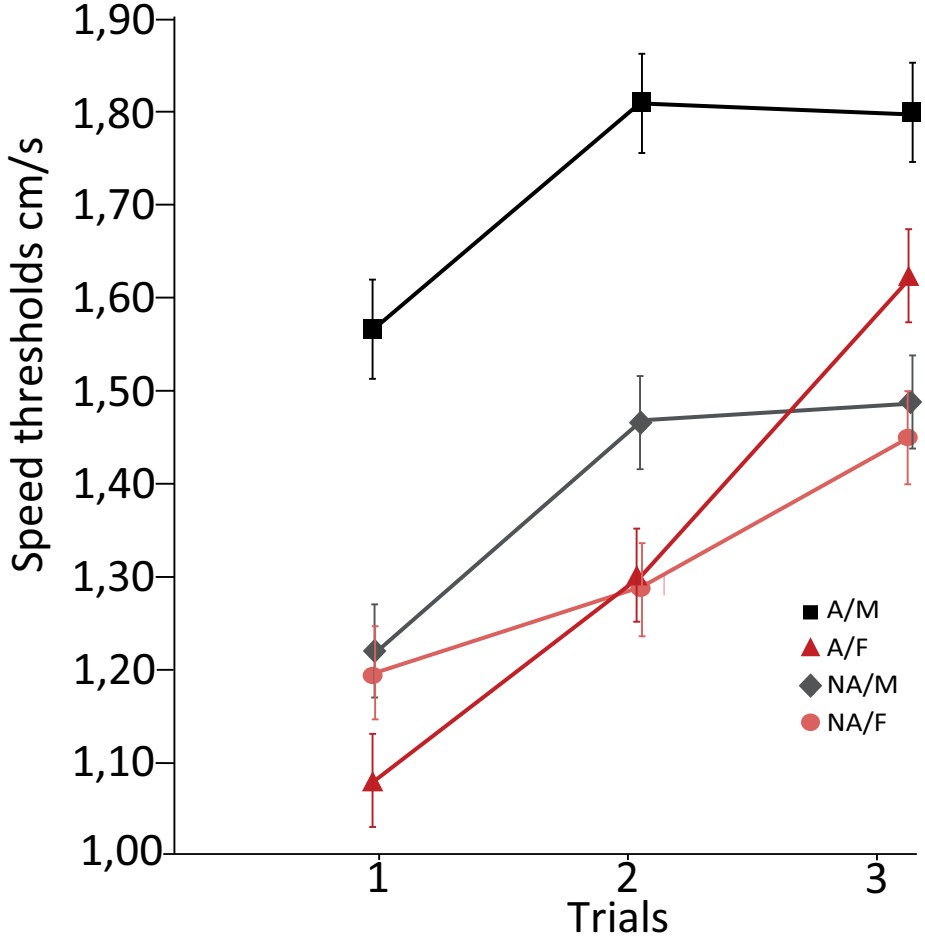

**Fig 2. Athletes and non-athletes speed thresholds for 3 trials.**

**Table 2. Paired t-test results for normalized data.**

| Groups | Normalized data |
|---|---|
| A/M-A/F | *t (1,18) = 2.152, p = 0.045* |
| A/F–NA/F | *t (1,18) = -2.594, p = 0.018* |
| A/M-NA/M | *t (1,19) = -0.211, p = 0.943* |
| NA/F-NA/M | *t (1,19) = -0.72, p = 0.835* |

A: Athletes / NA: Non-athletes / M: Male / F: Female.

and research demonstrating that athletes showed higher speed processing, reaction times and information processing [17,19]. Specifically, significant differences between athletes and non-athletes seem to be attributable to the performance of male athletes (see Fig 2), although female athletes demonstrated speed thresholds similar to those of nonathletes. Research on differences in speed thresholds between athletes and nonathletes have also revealed differences for male groups. However, none of these studies compared female performance.

Our results for young athletes are consistent with those reported by Roudaia and Faubert [29], where males showed a higher tracking speed than females. However, unlike in their study, this difference was not observed in the nonathlete groups in our study, where both obtained similar speed thresholds. Additionally, one research review article observed that differences exist in speed processing and attentional paradigms (both elements solicited by the NT) between athletes and nonathletes, and males showed larger effects, but most studies have observed only male athletes, with few females included in their sample, which limits the ability to perform gender comparisons [17].

The athletic performance level has been found to influence 3D-MOT scores, as clearly demonstrated by Faubert in 2013. An obvious difference was present among professional athletes, elite amateur athletes and nonathletes (who did not participate in collective sports). Therefore, this difference could explain the speed threshold variances obtained in our sports groups between males and females. Although all athletes included in our sample were part of a college sports team, the coaches of our participants confirmed that the female athletes were not at a similar athletic training level as the male athletes. Qui and colleagues [35] observed no significant difference between intermediate athletes and nonathletes. Thus, if we consider our female athlete group to be intermediate athletes, it is consistent that no difference was observed compared to nonathletes.

Table 1 shows a significant relative change for female athletes compared to the male athlete group (*t* (1,18) = 2.152, *p* = 0.045) from trial 3 to trial 1. The female performance curve increased steadily over the 3 trials, whereas this curve appeared to plateau after the 2nd trial in males. The speed thresholds obtained in the 3rd trial were similar to those in the 2nd trial in males. Although we observed a similar pattern for both female groups, this difference was significant only for female athletes. Sports can lead them to higher speed thresholds than nonfemale athletes. Overall, the female speed threshold performance was lower than that of males.

One limitation of our study was the number of trials that used the NT. As observed in Table 1, participants observed higher speed thresholds after 3 trials. However, more trials should be evaluated to obtain a more realistic picture of participant performance. Previous research has shown that training on the 3D-MOT speed task over several sessions (average 15 sessions) generates an increase in the speed threshold performance in different populations [21,25, 36]. Furthermore, there were a variety of athletic training levels between genders. We cannot confirm whether the difference between genders in our athlete groups is linked to the training level or the gender itself. Similar training-level teams are needed for further investigation.

As mentioned above, athletic training creates brain modifications resulting in benefits in attention processing, decision making, etc. [37]. On the other hand, these attentional processes can be trained and can potentially lead to better performance on the field [21]. By increasing attention processes, athletes can potentially become more performant. Alternatively, athletic training can benefit the plasticity for attention mechanisms. As was demonstrated by Faubert (2013) sport expertise has a dramatic impact on attentional capacities for processing dynamic scenes as measured by the 3D-MOT task [23]. Therefore, it may be beneficial for female athletes to increase their participation in sports activities. However, it remains to be determined at what level the gap seen between males and females in attention mechanisms responsible for the 3D-MOT can be closed by engaging in competitive sports.

## Conclusions

In conclusion, we have successfully shown that participation in competitive sports results in a difference in perceptual cognitive abilities in adolescents, more specifically in male athletes. Males and females showed different perceptual cognitive abilities.

## Supporting information

**S1 Raw data.**
(XLSX)

## Author Contributions

**Conceptualization:** Isabelle Legault, Jocelyn Faubert.

**Data curation:** Isabelle Legault.

**Formal analysis:** Isabelle Legault.

**Funding acquisition:** Isabelle Legault.

**Investigation:** Isabelle Legault, Dylan Sutterlin-Guindon.

**Methodology:** Isabelle Legault, Jocelyn Faubert.

**Project administration:** Isabelle Legault.

**Resources:** Jocelyn Faubert.

**Software:** Jocelyn Faubert.

**Supervision:** Isabelle Legault.

**Validation:** Isabelle Legault.

**Visualization:** Isabelle Legault.

**Writing – original draft:** Isabelle Legault, Dylan Sutterlin-Guindon.

**Writing – review & editing:** Isabelle Legault, Jocelyn Faubert.

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
