## [Decision Letter · Decision Letter 0]

14 Apr 2022

PONE-D-21-37053Perceptual cognitive abilities in young athletes: A gender comparison.PLOS ONE

Dear Dr. Legault,

Thank you for submitting your manuscript to PLOS ONE. After careful consideration, we feel that it has merit but does not fully meet PLOS ONE’s publication criteria as it currently stands. Therefore, we invite you to submit a revised version of the manuscript that addresses the points raised during the review process.

I received one review from an expert in the field, and I have evaluated the manuscript myself.  The reviewer lists some concerns that need to be addressed before your manuscript is suitable for publication.  I concur with the reviewer's concerns.  I encourage you to add a data analysis section at the end of the Method section where you explain what tests you plan to run.  From my own reading, I feel like the results need to be unpacked more so readers can more easily interpret them.  If you feel you can adequately address these concerns then I invite you to submit a revised manuscript along with a detailed response to each point made by the reviewer.

We look forward to receiving your revised manuscript.

Kind regards,

Darrell A. Worthy, Ph.D

Academic Editor

PLOS ONE

Journal Requirements:

Reviewers' comments:

Reviewer's Responses to Questions

**Comments to the Author**

1. Is the manuscript technically sound, and do the data support the conclusions?

Reviewer #1: Partly

2. Has the statistical analysis been performed appropriately and rigorously? 

Reviewer #1: No

3. Have the authors made all data underlying the findings in their manuscript fully available?

Reviewer #1: Yes

4. Is the manuscript presented in an intelligible fashion and written in standard English?

Reviewer #1: Yes

5. Review Comments to the Author

Reviewer #1: The topic is interesting. However, the authors are recommended to reinforce the rationale linked to the gender differences within the introduction section. IMO, the fact that few studies have been conducted is not sufficient.

The Reviewer believe that a specific paragraph dedicated to the data analysis employed is needed.

Here below some specific comments:

Line 82: please reinforce the rationale of the study, explaining why it should important including a gender comparison.

line 95: the authors are invited to provide their study hypothesis in line with the logical flow of the introduction.

line 99: please justify the sample size included

Moreover, report more details about participants? Anthropometric data?

A specific section for data analysis is needed. Please provide it.

line 171: redundant. Remove it

Provide also some practical information to the readers dealing with young athletes.

6. PLOS authors have the option to publish the peer review history of their article (what does this mean?). If published, this will include your full peer review and any attached files.

Reviewer #1: No

---

## [Author Response · Author response to Decision Letter 0]

17 May 2022

Dear Editor, 

We are resubmitting our paper entitled “Perceptual cognitive abilities in young athletes: A gender comparison.” to you with the corrections suggested by you and the reviewer. Details can be found in our point-by-point response to the comments. 

We hope that these corrections will satisfy your expectations. 

Best Regards,

---

## [Editor Report · Decision Letter 1]

15 Aug 2022

Perceptual cognitive abilities in young athletes: A gender comparison.

PONE-D-21-37053R1

Dear Dr. Legault,

We’re pleased to inform you that your manuscript has been judged scientifically suitable for publication and will be formally accepted for publication once it meets all outstanding technical requirements.

I think you have successfully addressed all the concerns raised in the review process.

Kind regards,

Darrell A. Worthy, Ph.D

Academic Editor

PLOS ONE
---

## [Editor Report · Acceptance letter]

18 Aug 2022

PONE-D-21-37053R1 

Perceptual cognitive abilities in young athletes: A gender comparison. 

Dear Dr. Legault:

I'm pleased to inform you that your manuscript has been deemed suitable for publication in PLOS ONE. Congratulations! Your manuscript is now with our production department. 

Kind regards, 

on behalf of

Dr. Darrell A. Worthy 

Academic Editor

PLOS ONE